# Class 1 integron-borne cassettes harboring $bla_{CARB-2}$ gene in multidrug-resistant and virulent *Salmonella* Typhimurium ST19 strains recovered from clinical human stool samples, United States

**Daniel F. M. Monte**[1,2], **Fábio P. Sellera**[3], **Ralf Lopes**[4], **Shivaramu Keelara**[2], **Mariza Landgraf**[1], **Shermalyn Greene**[5], **Paula J. Fedorka-Cray**[2], **Siddhartha Thakur**[2,6]*

1 Department of Food and Experimental Nutrition, Food Research Center, Faculty of Pharmaceutical Sciences, University of São Paulo, São Paulo, Brazil, 2 Department of Population Health and Pathobiology, North Carolina State University, College of Veterinary Medicine, Raleigh, North Carolina, United States of America, 3 Department of Internal Medicine, School of Veterinary Medicine and Animal Science, University of São Paulo, São Paulo, Brazil, 4 Department of Clinical Analyses, Toxicology and Food Science, School of Pharmaceutical Sciences of Ribeirão Preto, University of São Paulo, Ribeirão Preto, Brazil, 5 Department of Health and Human Services, Molecular Diagnostic and Epidemiology Laboratory Unit at State Laboratory of Public Health, Division of Public Health, Raleigh, North Carolina, United States of America, 6 Comparative Medicine Institute, North Carolina State University, Raleigh, North Carolina, United States of America

* sthakur@ncsu.edu

**Data Availability Statement:** All relevant data are within the paper file.

## Abstract

International lineages, such as *Salmonella* Typhimurium sequence type (ST) 19, are most often associated with foodborne diseases and deaths in humans. In this study, we compared the whole-genome sequences of five *S.* Typhimurium strains belonging to ST19 recovered from clinical human stool samples in North Carolina, United States. Overall, *S.* Typhimurium strains displayed multidrug-resistant profile, being resistance to critically and highly important antimicrobials including ampicillin, ticarcillin/clavulanic acid, streptomycin and sulfisoxazole, chloramphenicol, tetracycline, respectively. Interestingly, all *S.* Typhimurium strains carried class 1 integron (*intI1*) and we were able to describe two genomic regions surrounding $bla_{CARB-2}$ gene, size 4,062 bp and 4,422 bp for *S.* Typhimurium strains (HS5344, HS5437, and HS5478) and (HS5302 and HS5368), respectively. Genomic analysis for antimicrobial resistome confirmed the presence of clinically important genes, including $bla_{CARB-2}$, *aac(6')-Iaa*, *aadA2b*, *sul1*, *tetG*, *floR*, and biocide resistance genes (*qacEΔ1*). *S.* Typhimurium strains harbored IncFIB plasmid containing *spvRABCD* operon, as well as *rck* and *pef* virulence genes, which constitute an important apparatus for spreading the virulence plasmid. In addition, we identified several virulence genes, chromosomally located, while the phylogenetic analysis revealed clonal relatedness among these strains with *S. enterica* isolated from human and non-human sources obtained in European and Asian countries. Our results provide new insights into this unusual class 1 integron in virulent *S.* Typhimurium strains that harbors a pool of genes acting as potential hotspots for horizontal gene transfer providing readily adaptation to new surrounds, as well as being crucially

**Funding:** The whole genome sequencing work is supported by the National Institutes of Health/Food and Drug Administration under award number 5U 18FD006194-02. NO.

**Competing interests:** The authors have declared that no competing interests exist.

required for virulence *in vivo*. Therefore, continuous genomic surveillance is an important tool for safeguarding human health.

## Introduction

Non-typhoidal *Salmonella* (NTS) is one of the most important foodborne pathogens with unprecedented impact on global health [1]. Among NTS, *Salmonella enterica* subsp. *enterica* serovar Typhimurium represents a major threat, since its worldwide spread has been associated with a broad host range, which includes mostly humans and food-related sources [1, 2]. Besides that, the emergence of multidrug-resistant (MDR) *S. enterica* is another crucial aspect for food-related outbreaks globally, limiting our therapeutic options [3].

In addition to the high global burden of salmonellosis, extended-spectrum β-lactamase (ESBL)-producing *S. enterica* strains have been recognized as high-priority bacteria causing serious public health issue (https://www.who.int/news-room/detail/27-02-2017-who-publishes-list-of-bacteria-for-which-new-antibiotics-are-urgently-needed). Aside from this, the emergence of mobile genetic elements (MGEs), for instance, class 1 integrons play an essential role in the global spread of antimicrobial resistance [4, 5]. Another aspect to be considered is the wide range of virulence package that is typically associated with *Salmonella* Pathogenicity Islands (SPI), contributing to the infection process among diverse hosts [6–8]. In this context, while the surveys with genomic approach have helped in the development of mitigation strategies and clinical management, continuous active surveillance is urgently required.

Here, we describe the genomic characteristics of five MDR and virulent *S.* Typhimurium strains carrying the *bla*<sub>CARB-2</sub> gene recovered from clinical human stool samples in North Carolina, United States.

## Materials and methods

### Ethics approval and consent to participate

The human patients from whom *Salmonella* were recovered were completely anonymous and even after all the analysis and tests, the human sample remained anonymous. As such, the NC State IRB (FWA: 00003429) indicated the study research did not need IRB approval because it does not meet the definition for human subjects research.

### Bacterial strains and antimicrobial susceptibility testing

We conducted a genomic investigation on five clinical *S.* Typhimurium strains collected in 2014 in North Carolina, United States. The strains were subjected to phenotypic characterization using the microdilution panel susceptibility approach on Gram-negative Sensititre plates (CMV3AGNF and GNX2F, Trek Diagnostic Systems, OH, USA) following the interpretative criteria of Clinical and Laboratory Standards Institute [9, 10]. The MDR profile was defined as resistant to three or more classes of antimicrobials [11]. All *S.* Typhimurium strains underwent molecular screening for class 1 integron by PCR [12, 13] and were subsequently characterized by whole-genome sequencing (WGS) according to Pornsukarom et al. [14].

### Whole-genome sequencing and phylogenetic analysis

Libraries were prepared using the Nextera XT DNA sample preparation kit (Illumina, San Diego, CA), which were multiplexed and sequenced on MiSeq platform (Illumina, San Diego,

CA, USA) at a paired-end read (300 bp). Resulted raw sequence reads underwent a strict quality control, as well as we obtained the draft genomes by using default settings in CLC workbench 10.1.1 (Qiagen) as per Monte et al. [15]. The sequencing data were deposited in NCBI (PRJNA613764). For each strain, we uploaded the sequences into Center for Genomic Epidemiology (http://genomicepidemiology.org/) to detect multilocus sequence typing (MLST), resistome, plasmid incompatibility groups and *Salmonella* Pathogenicity Islands.

Virulome analyzes were performed by using default settings available in VFanalyzer [16]. Additionally, the genetic context of *bla*<sub>CARB-2</sub> and presence of virulence genes were investigated using BLASTn analysis against the non-redundant (NR) database and manually curated using Geneious v. 11.1.5 (Biomatters Ltd., Auckland, New Zealand).

For phylogenetic purpose, we reconstructed a maximum likelihood phylogenetic tree based on single nucleotide polymorphism (SNP) using default settings of CSI Phylogeny version 1.4 [17]. SNP tree was reconstructed with five genomes of *S*. Typhimurium from this study in addition to thirteen genomes retrieved from GenBank database. Additional genomes of *S. enterica* strains were chosen from different sources (human, camel, food, poultry, ovine, river, and dog) and countries, including USA (SAMN10863500 and SAMEA6514930), France (SAMN07734943), Scotland (SAMEA773504 and SAMEA773551), Denmark (SAMEA4349586), Ireland (SAMEA4825483), Switzerland (SAMN08936646), Germany (SAMEA6058372), Chile (SAMN14336901), China (SAMN09759463 and SAMN02844307), and Ethiopia (SAMN03577126).

## Results

### Antimicrobial susceptibility testing and class 1 integron detection

All the five strains were classified as MDR, displaying resistance to critically important antimicrobials including ampicillin (100%), ticarcillin/clavulanic acid (100%), and streptomycin (60%), as well as to highly important antimicrobials comprising sulfisoxazole (100%), chloramphenicol (100%), and tetracycline (60%) (Table 1). Moreover, intermediate resistance to doxycycline was detected in three strains (HS5344, HS5437 and HS5478), and in a single strain (HS5437) to ceftazidime. In addition, we confirmed the presence of class 1 integron in all *S*. Typhimurium strains.

**Table 1. Phenotypic and genomic features of *Salmonella* Typhimurium ST19 strains isolated from clinical human samples in United States.**

| Strain ID | Serotype | Source | R-type (MIC)* | Resistance genotype | Plasmids | ST | Accession number |
|---|---|---|---|---|---|---|---|
| HS5302 | Typhimurium (O5-) | Stool | FIS-AMP-TIM2 | *bla*<sub>CARB-2</sub>, *aac(6')-Iaa*, *sul1* | IncFIB(S), IncFII(S) | 19 | JAATJP000000000 |
| HS5344 | Typhimurium (O5-) | Stool | CHL-TET-FIS-AMP-STR-TIM2 | *bla*<sub>CARB-2</sub>, *aac(6')-Iaa*, *aadA2b*, *sul1*, *tet(G)*, *floR* | IncFIB(S), IncFII(S) | 19 | JAATGY000000000 |
| HS5368 | Typhimurium (O5-) | Stool | FIS-AMP-TIM2 | *bla*<sub>CARB-2</sub>, *aac(6')-Iaa*, *sul1* | IncFIB(S), IncFII(S) | 19 | JAATJO000000000 |
| HS5437 | Typhimurium (O5-) | Stool | CHL-TET-FIS-AMP-STR-TIM2 | *bla*<sub>CARB-2</sub>, *aac(6')-Iaa*, *aadA2b*, *sul1*, *tet(G)*, *floR* | IncFIB(S), IncFII(S) | 19 | JAATGZ000000000 |
| HS5478 | Typhimurium | Stool | CHL-TET-FIS-AMP-STR-TIM2 | *bla*<sub>CARB-2</sub>, *aac(6')-Iaa*, *aadA2b*, *aph(3')-Ia*, *sul1*, *tet(G)*, *floR* | IncFIB(S), IncFII(S) | 19 | JAATHA000000000 |

*FIS, sulfisoxazole; AMP, ampicillin; TIM2, ticarcillin/clavulanic acid constant 2; CHL, chloramphenicol; TET, tetracycline; STR, streptomycin.

## Whole-genome sequencing and phylogenetic analysis

Genomic analysis revealed that all five *S.* Typhimurium strains belonged to the international sequence type (ST) ST19, while antimicrobial resistome confirmed the presence of critically important genes, such as carbenicillinase [*bla*$_{CARB-2}$], aminoglycosides [*aac(6')-Iaa* and *aadA2b*], sulfonamide [*sul1*], tetracycline [*tetG*], and florfenicol [*floR*]. The IncFIB(S) and IncFII(S) plasmid incompatibility groups were detected in all strains. We were also able to describe two schematic representations of the genetic context surrounding *bla*$_{CARB-2}$ gene. First, three *S.* Typhimurium strains (HS5344, HS5437, and HS5478) analyzed in this study, shared a genomic environment with 4,062 bp in size composed by *groEL*/*intI1-bla*$_{CARB-2}$-*qacEΔ1-sul1-orf5* (acetyltransferase)-*orf6* (hypothetical protein). Second, the remaining *S.* Typhimurium strains (HS5302 and HS5368) presented a genomic content slightly different with a 4,422 bp region composed by *intI1-bla*$_{CARB-2}$-*qacEΔ1-sul1-orf5* (acetyltransferase)-*orf6* (hypothetical protein) (Fig 1). Additionally, the *sul1*, *bla*$_{CARB-2}$, *tetG*, *floR*, and *aadA2b* resistance genes were harbored by a partial sequence of a complex class 1 integron (In104) from HS5344, HS5437, and HS5478. This sequence included duplications of parts of the integron conserved segments (CS), specifically, part of the *intI1* gene from the 5'-CS and part of the 3'-CS (*qacEΔ1* and partial *sul1* genes). Consequently, the structure had two *attI1* sites, into which the *aadA2b* gene cassette was inserted in one and the *bla*$_{CARB-2}$ cassette in the other. The *floR* and *tetG* genes were identified between the two integron-derived regions. In HS5302 and HS5368, only the region containing the *intI1-bla*$_{CARB-2}$-*qacEΔ1-sul1-orf5-orf6* array was detected. Furthermore, while *aac(6')-Iaa* was found at a site distant from the other resistance genes on the chromosome of all *S.* Typhimurium strains in this study, *aph(3')-Ia* was identified in a partial transposon sequence from HS5478.

Virulome analysis revealed presence of several *Salmonella* Pathogenicity Island (SPI-1, SPI-2, SPI-3, SPI-4, SPI-5, SPI-13, SPI-14, and Centisome 63 Pathogenicity Island) as shown in Table 2. Upon encountering these SPI, we also identified important virulence genes involved in fimbrial adherence (*fimA, C, D, F, H, I, W, Y, Z*), non-fimbrial adherence (*misL*), invasion (*InvA, B, C, E, F, G, H, I, J*), secretion system (*ssa, ssc, sse,* and *ssr*), magnesium uptake (*mgtB* and *mgtC*), regulation (*phoP, phoQ,* and *pipB*), and translocated effector (*sopB/sigD* and *sopE2*) (Table 2).

Interestingly, these strains possess a highly conserved *spv* operon composed by *spvR, spvA, spvB, spvC,* and *spvD* genes which are located upstream of the genes *pefA* (plasmid-encoded fimbriae) and *rck* (resistance to complement killing) in a virulence plasmid as shown in Fig 2. In addition, *in silico* analyses confirmed that these virulence genes were located on IncFIB plasmid.

To achieve a better understanding of the clonal spread of these MDR strains, we reconstructed a phylogenetic tree based on SNPs. Indeed, these strains were found to be genetically related. The phylogenetic tree framed a major cluster composed by five *S.* Typhimurium strains from this study (HS5478, HS5344, HS5437, HS5302, and HS5368), which nested

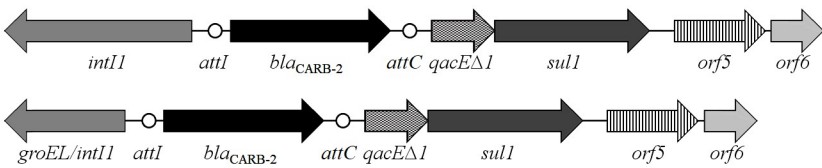

**Fig 1. Schematic representation of the genetic context surrounding *bla*$_{CARB-2}$ genes in *Salmonella* Typhimurium ST19 strains isolated from clinical human samples in United States.**

**Table 2. Genomic features of virulence factors of *Salmonella* Typhimurium ST19 strains isolated from clinical human samples in United States.**

| Strain ID | SPI-1 encode genes | SPI-2 encode genes* | SPI-3 encode genes | SPI-5 encode genes | Virulence plasmid* | Fimbrial adherence determinants | SPI* |
|---|---|---|---|---|---|---|---|
| HS5302 | *inv (A, B, C, E, F, G, H, I, J); sopE2* | *ssa (C, D, E, G, H, **I**, J, K, L, M, N, O, P, Q, R, T, U, V); ssc (A, B); sse (B, C, D, E); ssr (A, B)* | *mgtB, mgtC, misL* | *phoP, phoQ, pipB, sopB/ sigD* | *spv (A, B, D, R)* | *fim (A, C, D, F, H, I, W, Y, Z)* | SPI-1, SPI-2, SPI-3, SPI-5, SPI-13, SPI-14, C63PI |
| HS5344 | *inv (A, B, C, E, F, G, H, I, J); sopE2* | *ssa (C, D, E, G, H, J, K, L, M, N, O, P, Q, R, T, U, V); ssc (A, B); sse (B, C, D, E); ssr (A, B)* | *mgtB, mgtC, misL* | *phoP, phoQ, pipB, sopB/ sigD* | *spv (A, B, D, R)* | *fim (A, C, D, F, H, I, W, Y, Z)* | SPI-1, SPI-2, SPI-3, **SPI-4**, SPI-5, SPI-13, SPI-14, C63PI |
| HS5368 | *inv (A, B, C, E, F, G, H, I, J); sopE2* | *ssa (C, D, E, G, H, **I**, J, K, L, M, N, O, P, Q, R, T, U, V); ssc (A, B); sse (B, C, D, E); ssr (A, B)* | *mgtB, mgtC, misL* | *phoP, phoQ, pipB, sopB/ sigD* | *spv (A, B, D, R)* | *fim (A, C, D, F, H, I, W, Y, Z)* | SPI-1, SPI-2, SPI-3, **SPI-4**, SPI-5, SPI-13, SPI-14, C63PI |
| HS5437 | *inv (A, B, C, E, F, G, H, I, J); sopE2* | *ssa (C, D, E, G, H, **I**, J, K, L, M, N, O, P, Q, R, T, U, V); ssc (A, B); sse (B, C, D, E); ssr (A, B)* | *mgtB, mgtC, misL* | *phoP, phoQ, pipB, sopB/ sigD* | *spv (A, B, D, R)* | *fim (A, C, D, F, H, I, W, Y, Z)* | SPI-1, SPI-2, SPI-3, **SPI-4**, SPI-5, SPI-13, SPI-14, C63PI |
| HS5478 | *inv (A, B, C, E, F, G, H, I, J); sopE2* | *ssa (C, D, E, G, H, J, K, L, M, N, O, P, Q, R, T, U, V); ssc (A, B); sse (B, C, D, E); ssr (A, B)* | *mgtB, mgtC, misL* | *phoP, phoQ, pipB, sopB/ sigD* | *spv (A, B, D, R)* | *fim (A, C, D, F, H, I, W, Y, Z)* | SPI-1, SPI-2, SPI-3, SPI-5, SPI-13, SPI-14, C63PI |

*Letters highlighted in bold represents differences among strains.

together with *S. enterica* strains from different sources (Human, poultry, ovine) and countries, including Denmark (SAMEA4349586), Scotland (SAMEA773504 and SAMEA773551), France (SAMN07734943), and China (SAMN09759463) as shown in Fig 3. Interestingly, *S.* Typhimurium strains within same cluster shared the same resistance phenotype and genotype profile.

## Discussion

The continuous dispersal of MDR *S. enterica* strains frequently deserves attention of the public health authorities, particularly the international lineages as *S.* Typhimurium ST19 that most often causes diseases and deaths [18, 19]. Owing to their importance, the ST19 members have been globally identified in a variety of sources, such as human clinical samples, animals, food products, and environmental samples [20–22]. Moreover, *S.* Typhimurium ST19 has shown broad resistance to a variety of critically important antimicrobials [23], including colistin (an antibiotic of last resort for some MDR infections) [24, 25]. Besides that, the occurrence of intermediate resistance reported here implies in possible treatment failure that should be noted by public health authorities.

It is important to note that these strains can easily acquire such genes through mobile genetic elements such as plasmids, integrons, and genomic islands from other MDR clones, resulting in their rapid dissemination. The presence of class 1 integron in all *S.* Typhimurium ST19 strains constitutes a risk factor to the rapid spread of antimicrobial resistance (AMR) genes. Indeed, class 1 integron coding various resistance profiles has been widely reported in *S.* Typhimurium as well as in multiple serovars [5, 21, 26–30]. This genetic frame is crucial for the spread resistance markers, since they are able to capture AMR genes through chromosomal cassettes incorporating them by site-specific recombination [4, 12, 31]. Additionally, resistance genes located in class 1 integrons are often within *Salmonella* genomic islands (SGI), such as the conjugative *Salmonella* genomic island 1 (SGI1) (~43-kb) and its variants [32, 33].

The detection of quaternary ammonium compounds (QACs) raises a particular concern, since this *qac*-containing integrons typically harbors a pool of genes that are hotspots for

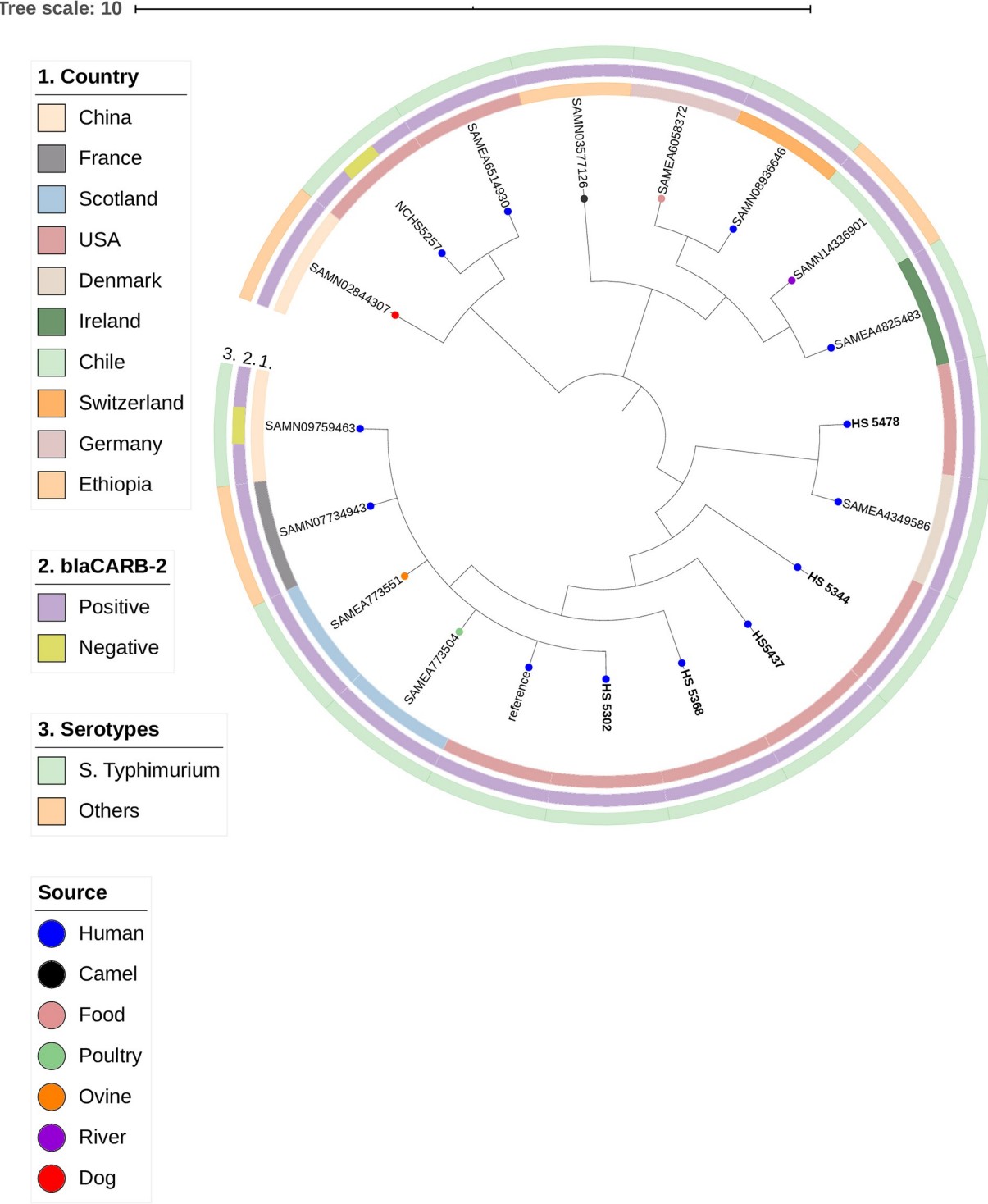

**Fig 2. Genomic comparison between genetic contexts of virulence plasmids carried by *Salmonella* Typhimurium strains from this study (A) and *S. enterica* strains B (CP000858), C (NC_002638), and D (AY517905) as out-group.** Genes and shotgun sequences were retrieved from the GenBank database. Arrows indicate the positions and directions of the genes; Regions with >99% identity are indicated with gray shading.

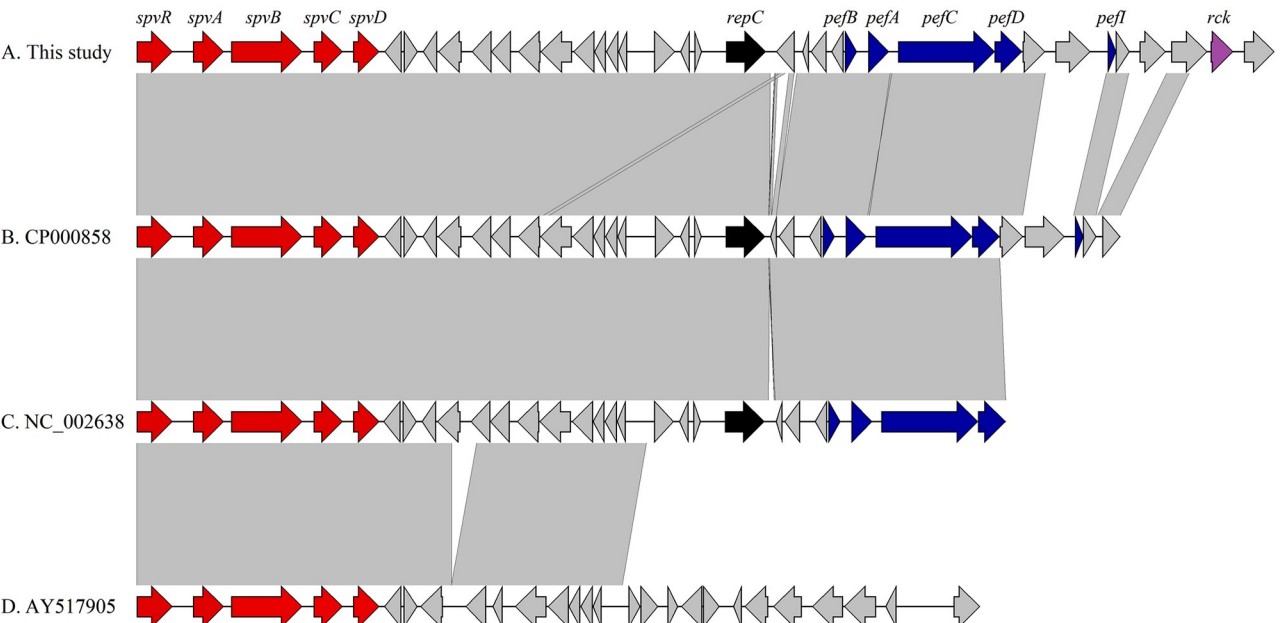

**Fig 3. SNP-based phylogenetic tree composed by five *Salmonella* Typhimurium and additional 14 *Salmonella enterica* strains.** This figure was generated with iTOL v.5.5 (https://itol.embl.de).

horizontal gene transfer providing readily adaptation to new surrounds [34, 35]. The co-resistance of critically important antimicrobials and disinfectants QACs reinforces the evidence of the overuse of biocides in clinical settings [34], and their spread have been also described in *Salmonella* serotypes isolated from livestock [36].

The *bla*$_{CARB-2}$ gene, earlier identified as *bla*$_{PSE-1}$, is most often a part of the chromosomal cassette [37, 38]. To date, the occurrence of this carbenicillinase gene has been limited to a few reports in different bacteria species and countries, including *Acinetobacter pittii* and *Salmonella* serovars in Australia [38, 39], *Salmonella* Typhimurium from England and Wales [40], *Salmonella* Senftenberg in Mexico [41], *S.* Typhimurium in Canada [42], *Pseudomonas aeruginosa* in Netherlands [43], and *Escherichia coli* in Pakistan [44]. It is noteworthy that such genetic element has the ability to move among different lineages of *S. enterica* serovars on a global scale, contributing to AMR spread [28]. Indeed, the genetic contexts surrounding *bla*$_{CARB-2}$ gene in this study are typically found in SGI1 and its variant SGI1-B.

Drug-resistant variants of SGI1 have been identified in numerous *S. enterica* serovars, and strains harboring them may be more virulent and have a tendency to rapidly disseminate [33, 39]. In fact, *S.* Typhimurium strains within this survey demonstrate to possess several virulence factors, which have been reported earlier [45–48]. Furthermore, we confirmed the presence of several plasmid-borne virulence genes (*spvR*, *spvA*, *spvB*, *spvC*, *spvD*, *rck*, and *pefA*) that denotes an important genomic apparatus for the spreading of this plasmid, and may provide fitness benefit as previously reported [28, 49, 50]. Increasing evidences have demonstrated that the *spv* operon affects the formation of autophagosomes, as well as highlight its association in killing of macrophages and neutrophils [6], being crucially required for virulence *in vivo* [8], including aggravated damage in zebrafish infection model [7]. Furthermore, the PhoP-regulated gene *mig-14* that is required for virulence and resistance to antimicrobial peptides was detected in these strains. Yet, *mig-14* contributes to *Salmonella* persistence in hosts, being also associated with resistance against polymyxin B and cathelin-related antimicrobial peptide

(CRAMP) [51–54]. Thus, the clonal dissemination of MDR *S.* Typhimurium (mostly the invasive clones) constitutes an important issue to public health [55], especially *S.* Typhimurium ST19, which have been circulating worldwide (http://enterobase.warwick.ac.uk/) as demonstrated in this study, since our *S.* Typhimurium strains nested with international lineages from at least four countries (Fig 2).

In summary, we report the genomic features of virulent and MDR *S.* Typhimurium ST19 strains carrying the *bla*$_{CARB-2}$ gene recovered from clinical human samples in United States. Our results provide new insights into this genetic environment that besides *bla*$_{CARB-2}$, contains genes, coding resistance to quaternary ammonium compounds (*qacEΔ1*) and sulfonamides (*sul1*). Furthermore, our findings could aid in understanding the epidemiology of *S.* Typhimurium ST19, which are of great value to initiate preventive measures to safeguard human health. Given the high spread of this international lineage, especially among the young and the elderly or immunocompromised people, public health authorities and regulatory food agencies need to be aware of the potential impact in public health and in economy caused by such pandemic MDR *S.* Typhimurium ST19 lineage, with particular attention in high-burden areas.

## Acknowledgments

We thank Lyndy Harden for generating the whole genome sequence profiles for the study.

## Author Contributions

**Conceptualization:** Daniel F. M. Monte, Siddhartha Thakur.

**Data curation:** Daniel F. M. Monte, Fábio P. Sellera, Ralf Lopes.

**Formal analysis:** Daniel F. M. Monte, Ralf Lopes.

**Funding acquisition:** Mariza Landgraf, Paula J. Fedorka-Cray, Siddhartha Thakur.

**Investigation:** Daniel F. M. Monte, Shermalyn Greene, Siddhartha Thakur.

**Methodology:** Daniel F. M. Monte, Shivaramu Keelara, Shermalyn Greene, Siddhartha Thakur.

**Supervision:** Siddhartha Thakur.

**Validation:** Daniel F. M. Monte.

**Visualization:** Daniel F. M. Monte.

**Writing – original draft:** Daniel F. M. Monte.

**Writing – review & editing:** Daniel F. M. Monte, Fábio P. Sellera, Ralf Lopes, Mariza Landgraf, Paula J. Fedorka-Cray, Siddhartha Thakur.

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
