## [Decision Letter · Decision Letter 0]

3 Sep 2020

PONE-D-20-21578

Class 1 Integron-Borne Cassettes harboring blaCARB-2 gene in Multidrug-Resistant Salmonella Typhimurium ST19 Strains recovered from clinical human stool samples, United States

PLOS ONE

Dear Dr. Thakur,

Thank you for submitting your manuscript to PLOS ONE. After careful consideration, we feel that it has merit but does not fully meet PLOS ONE’s publication criteria as it currently stands. Therefore, we invite you to submit a revised version of the manuscript that addresses the points raised during the review process.

Reviewer comments on the manuscript need to be addressed.

We look forward to receiving your revised manuscript.

Kind regards,

Iddya Karunasagar

Academic Editor

PLOS ONE

Journal Requirements:

Additional Editor Comments (if provided):

The reviewer comments available on the manuscript. These need to be addressed.

Reviewers' comments:

Reviewer's Responses to Questions

**Comments to the Author**

1. Is the manuscript technically sound, and do the data support the conclusions?

Reviewer #1: Yes

2. Has the statistical analysis been performed appropriately and rigorously? 

Reviewer #1: N/A

3. Have the authors made all data underlying the findings in their manuscript fully available?

Reviewer #1: Yes

4. Is the manuscript presented in an intelligible fashion and written in standard English?

Reviewer #1: Yes

5. Review Comments to the Author

Reviewer #1: Review Comments to the Author

Please use the space provided to explain your answers to the questions above. You may also include additional comments for the author, including concerns about dual publication, research ethics, or publication ethics. (Please upload your review as an attachment if it exceeds 20,000 characters) (Limit 200 to 20000 Characters)

Could you please see my attached review, I have added all my comments to the Author in an attachment.

6. PLOS authors have the option to publish the peer review history of their article (what does this mean?). If published, this will include your full peer review and any attached files.

Reviewer #1: No

---

## [Author Response · Author response to Decision Letter 0]

2 Oct 2020

Response to reviewers, Manuscript PONE-D-20-21578.R1

Line numbers in our response refer to the line numbers in the revised manuscript (marked-up).

General comments:

We sincerely appreciate the editorial board and reviewers for their careful review and constructive suggestions. It is our belief that the manuscript was substantially improved after making the suggested edits. We have responded to specific queries below.

Comments from Reviewers:

Reviewer #1:

This manuscript describes about the class 1 integron AMR cassettes harboring blaCARB-2 gene in MDR Salmonella Typhimurium ST19 isolates. These MDR isolates are also harbored several virulence encoding genes and SPIs.

Response: We sincerely thank the reviewer for the favorable comments. 

Comments:

Based on the WHO’s description, some of the antimicrobial such as ticarcillin/clavulanic acid and tetracycline are not critically and highly important antimicrobials.

Response: We appreciate your suggestion. However, based on this document: Critically important antimicrobials for human medicine, 6th revision. Geneva: World Health Organization; 2019. https://apps.who.int/iris/bitstream/handle/10665/312266/9789241515528-eng.pdf?sequence=1&isAllowed=y, we could observe that ticarcillin/clavulanic acid and tetracycline are classified as critically and highly important antimicrobials. 

Lines 69-70. ‘CARB-2-production’ was not tested in these S. Typhimurium isolates, but only the encoding gene was detected.

Response: We agree. This has been deleted in lines 72 and 245. 

Line 75. It will be useful if the nature of clinical source (watery diarrhea, diarrhea with mucus or dysentery-like stool) of the isolates and clinical history of the patients described and included. In addition, the year/location of isolation is also important to be mentioned in the text to deduce whether the isolates are spread all over the country or restricted to a certain geographical location.

Response: We fully agree. At the beginning of this study, we tried to collect such information to conduct the paper describing the clinical history of the patients. In our understanding, such information could aid to mitigate strategies, as well as give to readers a full background, which denotes the real threat of salmonellosis and their impacts in human health. However, we did not get permission from Public Health Laboratory due to health information privacy to publish this information. Therefore, we respectfully request that the paper be published with the information provided.

Lines 100-101. What is the rational for selecting only 13 sequences for comparison? There other STs (e.g., ST39) that expresses more AMR than the ST19. It will be interesting to consider all the MDR S. Typhimurium STs from different sources.

Response: Our intention was to cover several countries (n=10) and sources (n=7) in this phylogeny by using S. enterica strains harboring blaCARB-2 gene, with exception of two strains that were used as out-groups. 

Lines 114-115. It is not surprising, as these intermediate  resistant isolates to doxycycline are also resistant to tetracycline. Doxycycline is a synthetic antibiotic derived from tetracycline.

Response: We agree. This has been modified in lines 117-119. 

“Moreover, intermediate resistance to doxycycline was detected in three strains (HS5344, HS5437 and HS5478), and in a single isolate (HS5437) to ceftazidime”.

Lines 146-151. It will be interesting to compare these putative virulence encoding genes and SPIs with the infection status of the patients (pl refer comment #3).

Response: We agree. However, as prior mentioned, we are unable to provide this information. 

The title of the manuscript reflects only the MDR part, but not about the virulence.

Response: We agree. The title has been modified. “Class 1 Integron-Borne Cassettes harboring blaCARB-2 gene in Multidrug-Resistant and Virulent Salmonella Typhimurium ST19 Strains recovered from clinical human stool samples, United States”. (Lines 2-4).

There is inconsistency while using the terms isolate and strain.

Response: Thank you for this observation. Since the main target of this study was the genomic investigation and all Salmonella Typhimurium were accessed from a collection, we believe that the term strain is the most appropriate. Therefore, we have replaced these terms throughout the manuscript in lines 31, 33, 72, 76, 77, 78, 83, 114, 118, 120, 126, 163, 176, 237, and 245.

Only the blaCARB-2  was mapped in the class 1 integron. Are the other genes (aac(6’)-Iaa, aadA2b, aph(3’)-Ia, sul1, tetG, floR,) are located in the same AMR cassettes in the integron? How they are arranged on the chromosome?

Response: This information was added to the revised version of the manuscript. Lines 137-147: “Additionally, the sul1, blaCARB-2, tetG, floR, and aadA2b resistance genes were harbored by a partial sequence of a complex class 1 integron (In104) from HS5344, HS5437, and HS5478. This sequence included duplications of parts of the integron conserved segments (CS), specifically, part of the intI1 gene from the 5’-CS and part of the 3’-CS (qacE∆1 and partial sul1 genes). Consequently, the structure had two attI1 sites, into which the aadA2b gene cassette was inserted in one and the blaCARB-2 cassette in the other. The floR and tetG genes were identified between the two integron-derived regions. In HS5302 and HS5368, only the region containing the intI1-blaCARB-2-qacE∆1-sul1-orf5-orf6 array was detected. Furthermore, while aac(6’)-Iaa was found at a site distant from the other resistance genes on the chromosome of all S. Typhimurium strains in this study, aph(3’)-Ia was identified in a partial transposon sequence from HS5478.”

---

## [Decision Letter · Decision Letter 1]

7 Oct 2020

Class 1 Integron-Borne Cassettes harboring blaCARB-2 gene in Multidrug-Resistant and Virulent Salmonella Typhimurium ST19 Strains recovered from clinical human stool samples, United States

PONE-D-20-21578R1

Dear Dr. Thakur,

We’re pleased to inform you that your manuscript has been judged scientifically suitable for publication and will be formally accepted for publication once it meets all outstanding technical requirements.

Kind regards,

Iddya Karunasagar

Academic Editor

PLOS ONE

Additional Editor Comments (optional):

All reviewer comments addressed satisfactorily.

Reviewers' comments:

Reviewer's Responses to Questions

**Comments to the Author**

1. If the authors have adequately addressed your comments raised in a previous round of review and you feel that this manuscript is now acceptable for publication, you may indicate that here to bypass the “Comments to the Author” section, enter your conflict of interest statement in the “Confidential to Editor” section, and submit your "Accept" recommendation.

Reviewer #1: All comments have been addressed

2. Is the manuscript technically sound, and do the data support the conclusions?

Reviewer #1: Yes

3. Has the statistical analysis been performed appropriately and rigorously? 

Reviewer #1: N/A

4. Have the authors made all data underlying the findings in their manuscript fully available?

Reviewer #1: Yes

5. Is the manuscript presented in an intelligible fashion and written in standard English?

Reviewer #1: Yes

6. Review Comments to the Author

Reviewer #1: The questions above (1-5) are addressed adequately. I do not have any comments to the authors in this revised version.

7. PLOS authors have the option to publish the peer review history of their article (what does this mean?). If published, this will include your full peer review and any attached files.

Reviewer #1: No

---

## [Editor Report · Acceptance letter]

14 Oct 2020

PONE-D-20-21578R1 

Class 1 Integron-Borne Cassettes harboring *bla*_CARB-2_ gene in Multidrug-Resistant and Virulent *Salmonella* Typhimurium ST19 Strains recovered from clinical human stool samples, United States 

Dear Dr. Thakur:

I'm pleased to inform you that your manuscript has been deemed suitable for publication in PLOS ONE. Congratulations! Your manuscript is now with our production department. 

Kind regards, 

on behalf of

Dr. Iddya Karunasagar 

Academic Editor

PLOS ONE